# ePOCT+ and the medAL-*suite*: Development of an electronic clinical decision support algorithm and digital platform for pediatric outpatients in low- and middle-income countries

**Rainer Tan**[1,2,3,15]*, **Ludovico Cobuccio**[1,2,15], **Fenella Beynon**[2,15], **Gillian A. Levine**[2,15], **Nina Vaezipour**[2,15], **Lameck Bonaventure Luwanda**[3], **Chacha Mangu**[4], **Alan Vonlanthen**[5], **Olga De Santis**[1,6], **Nahya Salim**[3,7], **Karim Manji**[7], **Helga Naburi**[7], **Lulu Chirande**[7], **Lena Matata**[2,3,15], **Method Bulongeleje**[8], **Robert Moshiro**[7], **Andolo Miheso**[9], **Peter Arimi**[10], **Ousmane Ndiaye**[11], **Moctar Faye**[11], **Aliou Thiongane**[11], **Shally Awasthi**[12], **Kovid Sharma**[13], **Gaurav Kumar**[2,15], **Josephine Van De Maat**[14], **Alexandra Kulinkina**[2,15], **Victor Rwandarwacu**[2,15], **Théophile Dusengumuremyi**[2,15], **John Baptist Nkuranga**[16], **Emmanuel Rusingiza**[17,18], **Lisine Tuyisenge**[17], **Mary-Anne Hartley**[19], **Vincent Faivre**[5], **Julien Thabard**[5], **Kristina Keitel**[2,15,20]☯, **Valérie D'Acremont**[1,2,15]☯

**1** Digital and Global Health Unit, Unisanté, Centre for Primary Care and Public Health, University of Lausanne, Lausanne, Switzerland, **2** Swiss Tropical and Public Health Institute, Basel, Switzerland, **3** Ifakara Health Institute, Dar es Salaam, United Republic of Tanzania, **4** National Institute of Medical Research–Mbeya Medical Research Centre, Mbeya, United Republic of Tanzania, **5** Information Technology & Digital Transformation sector, Unisanté, Center for Primary Care and Public Health, University of Lausanne, Switzerland, **6** Institute of Global Health, University of Geneva, Geneva, Switzerland, **7** Department of Pediatrics and Child Health, Muhimbili University Health and Allied Sciences (MUHAS), Dar es Salaam, United Republic of Tanzania, **8** PATH, Dar es Salaam, United Republic of Tanzania, **9** PATH, Nairobi, Kenya, **10** College of Health Sciences, University of Nairobi, Nairobi, Kenya, **11** Department of Pediatrics, Cheikh Anta Diop University, Dakar, Senegal, **12** Department of Pediatrics, King George's Medical University, Lucknow, India, **13** PATH, Lucknow, India, **14** Radboudmc, Department of Internal Medicine and Radboudmc Center for Infectious Diseases, Nijmegen, Netherlands, **15** University of Basel, Basel, Switzerland, **16** Department of Paediatrics, King Faisal Hospital, Kigali, Rwanda, **17** University Teaching Hospital of Kigali, Kigali, Rwanda, **18** School of Medicine and Pharmacy, University of Rwanda, Kigali, Rwanda, **19** Intelligent Global Health, Machine Learning and Optimization Laboratory, Swiss Federal Institute of Technology (EPFL), Lausanne, Switzerland, **20** Paediatric Emergency Department, Department of Pediatrics, University Hospital Berne, Berne, Switzerland

☯ These authors contributed equally to this work.

* rainer.tan@unisante.ch

**Data Availability Statement:** The data that supports the findings outlined in supplementary

## Abstract

Electronic clinical decision support algorithms (CDSAs) have been developed to address high childhood mortality and inappropriate antibiotic prescription by helping clinicians adhere to guidelines. Previously identified challenges of CDSAs include their limited scope, usability, and outdated clinical content. To address these challenges we developed ePOCT+, a CDSA for the care of pediatric outpatients in low- and middle-income settings, and the medical algorithm suite (medAL-*suite*), a software for the creation and execution of CDSAs. Following the principles of digital development, we aim to describe the process and lessons learnt from the development of ePOCT+ and the medAL-*suite*. In particular, this

material 3 is publicly available from Zenodo: DOI: 10.5281/zenodo.400380.

**Funding:** This work took place within the framework of the DYNAMIC project that is funded by the Fondation Botnar, Switzerland (grant n˚ 6278) as well as the Swiss Development Cooperation (grant n˚7F-10361.01.01) received by VDA. The TIMCI project funded by UNITAID (grant n˚2019-35-TIMCI) received by VDA allowed for adapting of ePOCT+ or the medAL-suite software to Senegal, Kenya and India. The funders had no role in study or software design, data collection and analysis, decision to publish, or preparation of the manuscript.

**Competing interests:** The authors have declared that no competing interests exist.

work outlines the systematic integrative development process in the design and implementation of these tools required to meet the needs of clinicians to improve uptake and quality of care. We considered the feasibility, acceptability and reliability of clinical signs and symptoms, as well as the diagnostic and prognostic performance of predictors. To assure clinical validity, and appropriateness for the country of implementation the algorithm underwent numerous reviews by clinical experts and health authorities from the implementing countries. The digitalization process involved the creation of medAL-*creator*, a digital platform which allows clinicians without IT programming skills to easily create the algorithms, and medAL-*reader* the mobile health (mHealth) application used by clinicians during the consultation. Extensive feasibility tests were done with feedback from end-users of multiple countries to improve the clinical algorithm and medAL-*reader* software. We hope that the development framework used for developing ePOCT+ will help support the development of other CDSAs, and that the open-source medAL-*suite* will enable others to easily and independently implement them. Further clinical validation studies are underway in Tanzania, Rwanda, Kenya, Senegal, and India.

## Author summary

In accordance with the principles of digital development we describe the process and lessons learnt from the development of ePOCT+, a clinical decision support algorithm (CDSA), and medAL-suite, a software, to program and implement CDSAs. The clinical algorithm was adapted from previous CDSAs in order to address challenges in regards to the limited scope of illnesses and patient population addressed, the ease of use, and limited performance of specific algorithms. Clinical algorithms were adapted and improved based on considerations of what symptoms and signs would be appropriate for primary care health workers, and how well these clinical elements predic a particular disease or severe outcome. We hope that by sharing our multi-stakeholder approach to the development of ePOCT+, it can help others in the development of other CDSAs. The medAL-*creator* software was developed to allow clinicians without IT programming experience to program the clinical algorithm using a drag-and-drop interface, intended to allow a wider range of health authorities and implementers to develop and adapt their own CDSA. The medAL-*reader* application, deploys the algorithm from medAL-*creator* to end-users following the usual healthcare processes within a consultation.

## Introduction

Electronic clinical decision support algorithms (CDSAs) have been implemented in low- and middle-income countries (LMICs) in order to address excessive mortality due to poor quality of health care [1], and antimicrobial resistance due to inappropriate antibiotic prescription [2–5]. Such tools provide guidance through every step of the outpatient consultation to ultimately suggest the diagnosis and management plan based on the entered symptoms, signs and test results [6]. CDSAs have shown to help clinicians better adhere to guidelines [7–9], which resulting in improved quality of care and, for some, more rational antibiotic prescription [10,11]. This has led the World Health Organization (WHO) and its Member States to prioritize the scale-up of digital health technologies [12,13].

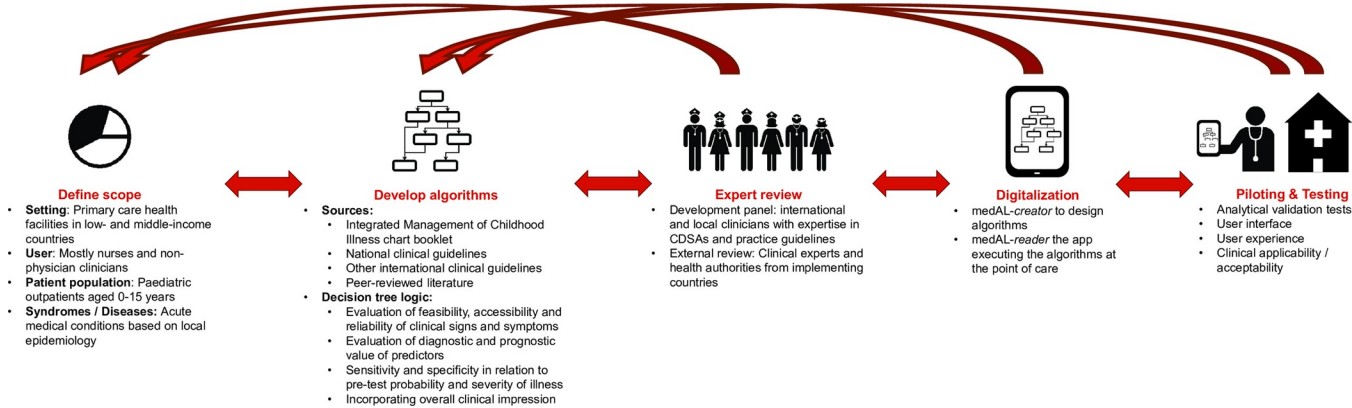

**Fig 1. Overall development process of ePOCT+ requiring multiple feedback loops.** The development process of ePOCT+ was an iterative process. We first defined the scope, then developed the algorithm (decision tree logic), followed by expert review with relevant stakeholders, the digitalization, and finally piloting and testing. Each stage resulted in multiple feedback loops to refine the end product.

Current CDSAs are not standardized, and concerns have been raised about their limited demographic and clinical scope [14,15], their usability [15,16], and their static and generic logic based on outdated guidelines that are unable to adapt to new evidence, evolving epidemiology, or changing resources. These challenges may contribute to variable uptake of CDSAs [16–18], and suboptimal performance when implemented [9,19].

In order to address these challenges, and build on the experience of previous CDSAs by our group [10,11], and others [6,9], we developed the CDSA ePOCT+, and a supporting digital software to create and execute CDSAs, the medAL-*suite*. ePOCT+ is currently being implemented in over 200 health facilities within the context of implementation studies in Tanzania, Rwanda, Senegal, Kenya and India. Following the principles of digital development and guidance on CDSAs [20–22], we aim to transparently share the rationale, strategy, and lessons learnt from this development process (Fig 1).

## Methods

### Scope

Compared to our previous generation CDSAs [6,10,11], the target level of care (primary health care facilities), and target users (mostly nurses and non-physician clinicians) remain the same. However, the target patient population was expanded from 2 months to 5 years, to also cover young infants below 2 months, and in some countries children 5 years up to 15 years.

The expanded target population age group adds young infants (<2 months) who are at highest risk of mortality [23], and children aged 5–15 years who are often neglected in international and national policies resulting in a slower decrease in mortality in LMICs compared to children under 5 years [24]. This expanded age group may help address the challenge of uptake by avoiding the need for clinicians to change tools when managing children of different age groups.

The scope of illnesses covered was also expanded in response to the frustration of clinicians using CDSAs who were not able to reach specific illnesses [14,16]. Expanding the scope allowed for the integration of common illnesses covered by other national clinical guidelines to which clinicians are expected to adhere, and to provide more opportunity for antibiotic stewardship when providing management guidance for specific illnesses.

Three major criteria were considered when expanding the scope of illnesses: 1) Incidence of presenting symptoms and diagnoses; 2) Morbidity, mortality, and outbreak potential; and 3) Capacity to diagnose and manage specific conditions at the primary care level.

Additional conditions were identified through: 1) national guidelines; 2) fever aetiology studies; 3) national health surveys; 4) chief complaints from primary care outpatient studies; 5) clinical expert review teams from the implementation countries; 6) interviews with end user clinicians; and 6) observation of consultations at primary health care facilities (Table A in S1 Appendix). Examples of notable additions for the Tanzanian algorithm include trauma, urinary tract infection, and abdominal pain that can account for 4.3–21.6% [25], 5.9–19.7% [25–27], and 4.6–23% [11,26] of outpatient consultations respectively.

## Clinical algorithm

The target users (mostly nurses and non-physician clinicians), and setting (primary health care facilities) were important considerations when identifying the guidelines and evidence to develop the algorithm. Previously validated algorithms [11], and the WHO Integrated Management of Childhood Illnesses (IMCI) chart booklets formed the backbone of the algorithm [28]. To support the expanded clinical scope, we turned to national guidelines to ensure adaptation to the local epidemiology, resources, and setting. If there was not sufficient detail in order to derive decision logic from these national guidelines, a brief review of literature was conducted to identify peer-reviewed literature and other international guidelines.

In order to transform narrative guidelines into Boolean decision tree logic algorithms, considerable interpretation was needed. The guiding principles for this process were derived from the properties to consider in the screening and diagnosis of a disease by Sackett and colleagues [29], the target product profile (TPP) for CDSAs as defined by experts in the field [21], and guidance on appropriate diagnostic and prognostic model development [30]. These include consideration of: a) the feasibility, acceptability, and reliability of clinical elements assessed at the primary care level, b) the diagnostic and prognostic value of individual and combined predictors, c) the sensitivity and specificity in relation to the severity and pre-test probability of the condition in the target population, and d) the overall clinical impression of the patient by the clinician.

a) Feasibility, acceptability, and reliability of predictors

If clinical algorithms are to be adequately utilized, the signs and symptoms used to reach a diagnosis must be feasible, acceptable and reliable when assessed by end-users. These properties were evaluated based on the results of several assessments: primarily an international Delphi study on predictors of sepsis in children [31], a systematic review on triage tools in low-resource settings [32], signs and symptoms included in established guidelines for primary health care workers such as IMCI [28], interviews with clinicians, observation of routine consultations, a Delphi survey among 30 Tanzanian health care workers (S2 Appendix), as well as subsequent feasibility tests observing clinicians using the CDSA on real and fictional cases. Notable findings from this process led to us not adding a pain score, capillary refill time, the assessment of cool peripheries, and weak and fast pulse, as they were deemed neither feasible nor reliable to be assessed at the primary care level. Importantly, these symptoms and signs are also not included within IMCI, likely for similar reasons [28].

b) Diagnostic and prognostic value of predictors

In the absence of validated diagnostic models for each diagnosis, we assessed individual diagnostic and prognostic factors to help guide the development of ePOCT+. Diagnostic studies derived from the population and setting of interest were preferred [33,34], as those developed from other settings often perform worse [35]. However, diagnostic predictors notably those predicting 'serious bacterial infection', often have low sensitivity, lack reference tests to

confirm bacterial origin, and ignore serious infections caused by viral diseases [36,37]. Prognostic studies are often better suited to develop clinical algorithms in order to understand which children are at risk of developing severe disease, regardless of the aetiology, to improve patient outcomes and reduce resource misallocation [38–40]. A systematic review of predictors of severe disease in febrile children presenting from the community helped identify useful clinical feature to be integrated within ePOCT+ [35], however few studies occurred at the primary care level. To address this gap we performed an exploratory analysis of clinical elements used in two CDSAs evaluated in Tanzania to predict clinical failure (S3 Appendix). This analysis found IMCI danger signs, severe general appearance, mid-upper arm circumference <12.5cm, oxygen saturation <90%, respiratory distress, and signs of anaemia and dehydration to be good predictors of clinical failure. Specific subgroup analyses on our previous generation CDSA provided further support for maintaining or modifying specific algorithm branches, particularly the inclusion of C-reactive Protein (CRP) point-of-care tests that helped safely reduce antibiotic prescription and improve confidence in management [41,42].

c) Sensitivity and specificity of algorithm branches in relation to severity and pre-test probability of condition

When constructing the algorithm, it was important to first identify children presenting with a severe condition, and only then use more specific branches to distinguish conditions requiring specific treatment from self-limiting illnesses requiring only supportive care (Fig 2). Predictors of severe conditions need to be sufficiently sensitive to guide interventions to

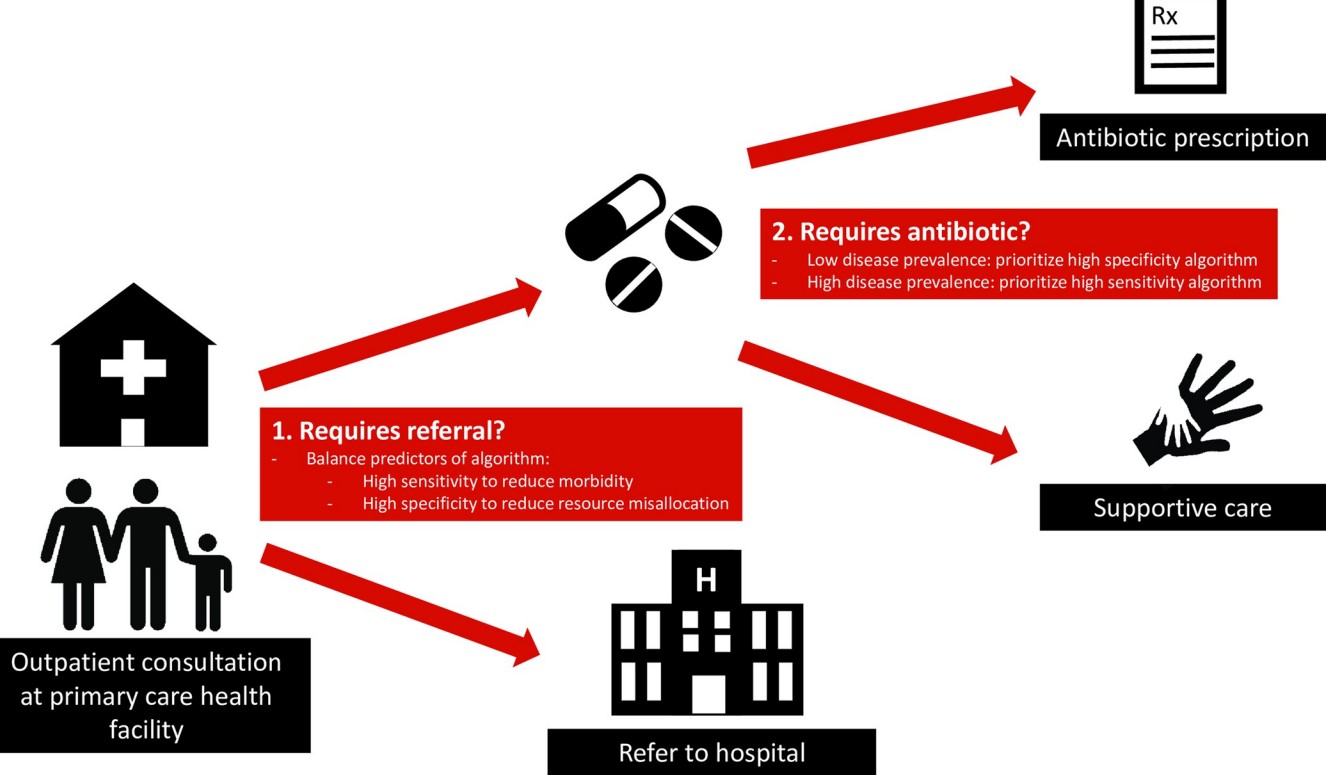

**Fig 2. Considering algorithm performance in regards to pre-test probability (disease prevalence) of the condition.** Health care workers are confronted with two major questions at primary care health facilities: 1) Does the child need to be referred? For which an algorithm must evaluate sensitivity and specificity in relation to the severity of disease. 2) Does the child require specific treatment (most often an antibiotic)? For which the disease prevalence of a bacterial illness needs to be considered when evaluating the sensitivity and specificity of such an algorithm.

reduce morbidity and mortality. However if this high sensitivity comes at the cost of reduced specificity, it can result in over-referral, misallocation of limited health care resources, and excess antibiotic prescription [38]. While this concept was considered within the development of the algorithm, most predictors and models studied lacked sufficient sensitivity and specificity to appropriately meet these requirements at the primary care level, thus emphasizing the need for better predictors and models [35,38].

Once a severe condition has been excluded, restricting antimicrobial prescriptions can be more safely integrated given the lower risk of clinical failure. Understanding the pre-test probability (disease prevalence) of the disease guides us on the level of specificity needed for the corresponding predictors to be included in the algorithm. In the outpatient settings, few non-severe children above 2 months have a condition requiring antibiotics [11,27]. As such, using the principles of Bayes' theorem [43], an algorithm for a condition of low prevalence requires a higher likelihood ratio to have a similar post-test probability than a condition with a higher prevalence. Within ePOCT+, C-Reactive Protein (CRP) test is integrated in several branches of the algorithm to increase specificity/likelihood ratio when the pre-test probability of requiring antibiotics is low. However, the pre-test probability of requiring antibiotics may increase in a child with comorbidities, and therefore a lower CRP cut-off can be used to increase sensitivity and reach the same post-test probability.

d) Integrating overall clinical impression

The overall clinical impression of a healthcare worker plays an important part in the diagnostic process [44], and may sometimes better identify serious conditions compared to isolated symptoms and signs [45,46]. As blindly following CDSA recommendations runs the risk of neglecting nuanced clinical observations or patient-initiated elements, we incorporated clinical impression in the algorithm to better preserve these skills [47]. More generally, it also shows a respect and consideration for the clinician's judgment and allows the tools to be more participatory; including the clinician in the interpretation and responsibility of the decision. As such, attempts were made to combine multiple clinical elements into one question utilizing clinical impression. This approach was used to help identify children who need a referral or antibiotics, such as "Severe difficult breathing needing referral", a criteria similar to that proposed by the British Thoracic Society [48], and "well/unwell appearing child", often used in children with fever without apparent source [36,49]. Highlighting in the application that this response will result in a recommendation of referral, aims to help clinicians understand the impact of their selection, and thus improve both the sensitivity and specificity. Such composite elements reduce the number of questions prompted by the CDSA, and speeds up the consultation process; an important consideration for uptake. Nevertheless, the diagnostic and prognostic value of the overall clinical impression of primary care clinicians in LMIC settings is not well understood, and further research is needed to understand how helpful these types of elements are when integrated within ePOCT+.

## Adapting and validating the medical content

ePOCT+ was first developed for Tanzania, where the prior generation of the algorithm was validated in a randomized-controlled trial [11]. Following the expansion and adaptation of the content described above, the algorithm was internally reviewed by 13 clinicians from 6 medical institutions with good understanding of CDSAs; 5 working in Tanzania, and the other 8 with experience working in LMICs. The ePOCT+ algorithm for Rwanda, Senegal, Kenya and India were then each drafted, with rounds of internal review, by small development teams composed of clinical algorithm development specialists, and national child health experts based on country-specific objectives, guidelines, and epidemiology, using the first algorithm as a scaffold.

In each country, the ePOCT+ algorithm was reviewed by a technical panel from the Ministry of Health or an independent clinical expert group (usually with Ministry of Health representatives). The panels were asked to assess the algorithm in terms of clinical validity, feasibility in primary care, scope of illnesses, and consistency with national policy and guidelines. The process of validation varied slightly in each country according to national decision-making mechanisms, but all included written feedback, individual and group meetings.

Certain algorithm branches were highlighted for group discussion; especially those with novel content, those for which significant interpretation was required from national guidelines, and any branches with queries or comments from panel members. For the algorithms with more novel content, more formal decision processes were used. In Tanzania and Rwanda a modified nominal group method was used, in which each participant one-by-one provided their opinion on the presented branch of the algorithm, followed by a group discussion and then an absolute majority vote for the final version.

Following the internal and external reviews, further modifications were made during the digitalization process, and feasibility tests, including feedback and review from end-users. For each proposed major change, the modification was communicated to the group to allow subsequent feedback and final approval by health authorities.

## Digitalization of ePOCT+ and development of the medAL-suite

We performed a landscaping review of existing CDSA software with respect to user interface, open source, data management, ease of programming and interpretation of clinical algorithms, and operability in target health facilities. Since none of the available software packages met our requirements, we developed the medAL-suite software following the requirements of the target product profile for CDSAs [21]. medAL-*creator* allows clinical experts to design the clinical content and logic of the algorithm, while medAL-*reader* is an Android based interface to execute the algorithm to end-user clinicians (Fig 3). Both software were developed collaboratively between the clinicians, IT programmers, end-users via feedback from field tests, and health authorities from the implementation countries.

The World Health Organization (WHO) have recently proposed the SMART guidelines to provide guidance and structure to translate the narrative guidelines (Layer 1), to semi-structured "human readable" decision trees and digital adaptation kits (Layer 2), to computer/machine readable structured algorithms (Layer 3), to the executable form of the software (Layer 4), and finally dynamic algorithms that are trained and optimised to local data (Layer 5) [50]. Each "translation" between layers is prone to interpretation and error, especially when each layer is developed by different actors and continuously adapted. To reduce error in interpretation, a major feature of medAL-*creator* is to allow the "computer/machine readable" structured algorithms to be "human readable", thus merging Layers 2 and 3. medAL-*creator* features a "drag and drop" user interface and automatic terminology/code set enabling the clinicians with no programming knowledge to create and review the algorithm. medAL-*reader* is then able to automatically convert the algorithm from medAL-*creator* for use at point-of-care.

medAL-*reader*, was designed based on our previous experiences of CDSA interfaces [8,11], and expert guidance on successful strategies in order for the application to be intuitive to use with limited training, to align with normal workflows at primary health care facilities, and encourage user autonomy [21,51,52].

## Validation tests and user-experience evaluations

Validation tests were performed for each diagnosis to ensure that the inputted data within medAL-*creator* were processed correctly into the expected output on medAL-*reader*. This

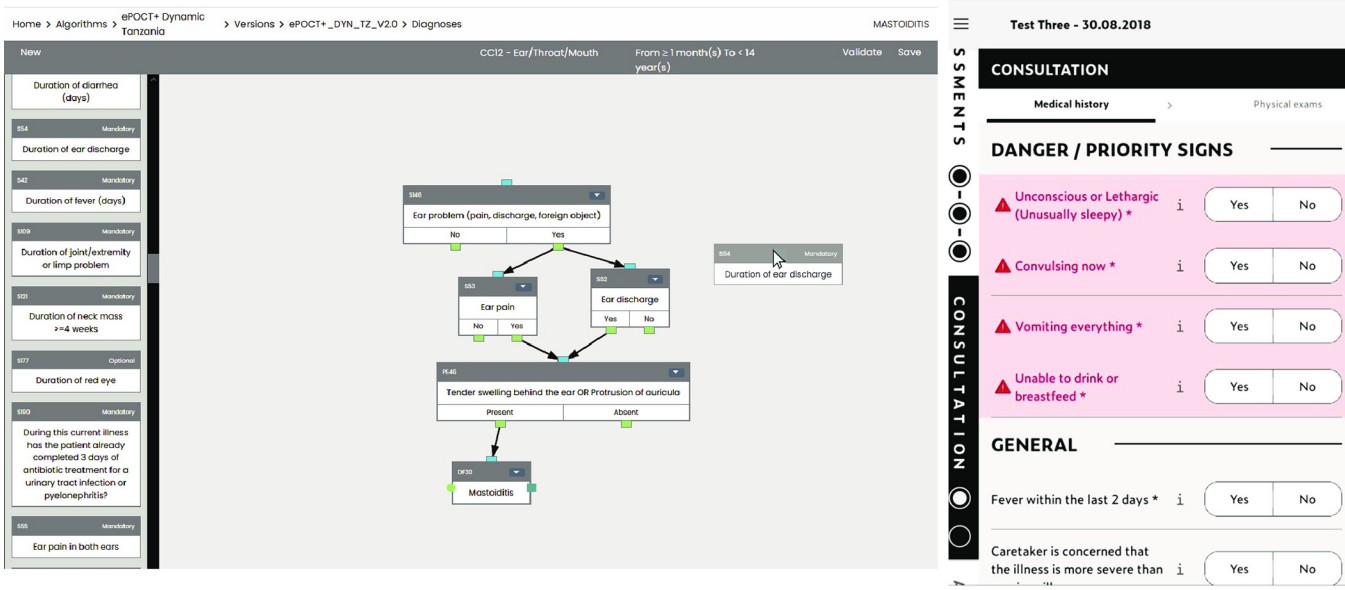

A) medAL-*creator*

B) medAL-*reader*

**Fig 3. medAL-*creator* and medAL-*reader*.** A) medAL-*creator* and its "drag and drop" user interface to design the clinical algorithm. For each clinical element a description and/or photo can be included to assist the end-user using medAL-*reader*; B) medAL-*reader* the android based application to collect the medical history, exposures, symptoms, signs and tests, and then propose the appropriate diagnosis and management.

included automated unit and integration testing, as well as automated non-regression testing by medAL-*creator*, and manual verification of medication posology for all drugs according to weight and age of the patient. All issues were reviewed by a clinical and IT team to correct the problems. While such tests are encouraged by the CDSA TPP [21], since CDSAs are not considered a "software as a medical device" by the Food and Drug Administration (FDA) [53] or European Medical Device Coordination Group [54], these tests are not legally required.

The ePOCT+ tool underwent numerous types and rounds of testing. To start, over 500 desk-based review cases focusing on user interface and analytical validation were performed by the various team members. Analytical validation tests ensured that the clinical content that was programmed in medAL-*creator* had the correct output in the medAL-*reader* application. End-user testing using fictional cases and supervised consultations concentrated on user experience, acceptability, and clinical applicability. Finally integrated testing in real-life conditions were performed where feedback was sought regularly. All user experience feedback was reviewed by a team including both clinical and IT specialists, while all clinical content modifications were approved by both the internal and external review panels.

## Ethics

Activities related to the development and piloting of ePOCT+ and the medAL-suite were done within the studies of DYNAMIC and TIMCI, for which approval was given from each country of implementation. The study protocol and related documents were approved by the institutional review boards of the Ifakara Health Institute in Tanzania (IHI/IRB/No: 11–2020 and 49–2020), the National Institute for Medical Research in Tanzania (NIMR/HQ/R.8a/Vol. IX/ 3486 and NIMR/HQ/R.8a/Vol. IX/3583), the National Ethics Committee of Rwanda (752/ RNEC/2020), the Comité National d'Ethique pour la Recherche en Santé of Senegal (SEN20/ 50), the University of Nairobi Ethics and Research Committee in Kenya (UON/CHS/TIMCI/

1/1), the King George's Medical College Institutional Ethics Committee in India (103rd ECM IC/P2), the Indian Council of Medical Research (2020–9753), the cantonal ethics review board of Vaud, Switzerland (CER-VD 2020–02800 & CER-VD 2020–02799), and the WHO Ethics Review Committee (ERC.0003405 & ERC.0003406). Written informed consent was obtained from all parents or guardians of children involved in the piloting of ePOCT+ and medAL-*reader*. No informed consent was obtained from health care workers involved in the development and refinement of the tools.

The exploratory analysis of predictors from the 2014 ePOCT study received approval of the study protocol and related documents by the institutional review boards of the Ifakara Health Institute and the National Institute for Medical Research in Tanzania (NIMRrHQ,R.8a,/trI'-VoIl. 789), by the Ethikkommission Beider Basel in Switzerland (EKNZ UBE 15/03), and the Boston Children's Hospital ethical review board. Written informed consent was obtained from all parents or guardians.

## Results

The ePOCT+ clinical algorithm and supporting evidence for each country of implementation can be found on the websites of the DYNAMIC and TIMCI studies that are implementing ePOCT+. The major features of medAL-*creator* and medAL-*reader* are summarized in the supplementary material (S4 Appendix), including the requirements defined by the CDSA target product profile (S5 Appendix).

The feasibility tests of ePOCT+ were conducted in over 200 patients in 20 health facilities, leading to numerous modifications (Table 1). The improved algorithm was then piloted with over 2000 consultations following 2 days of training and on-site support, before officially starting the clinical validation studies in the five countries of implementation.

## Discussion

ePOCT+ was derived from existing evidence and clinical validation field studies from previous generation CDSAs [8,10,11]. Novel content in the algorithm compared to other CDSA include

**Table 1. Example of modifications based on user-experience feedback and observations.**

| Issue | Description + context | Modifications |
|---|---|---|
| CDSA impractical in emergency situations | Child with convulsions was brought into the consultation room interrupting the current consultation. The clinician stopped using the tablet and managed the child providing the incorrect antibiotic class and dose | Emergency button integrated so that emergency management guidance can easily be accessed at any point of the algorithm. |
| Understanding algorithm branches | Why a patient reached a specific diagnosis was not always well understood by clinicians | To improve understanding, and to have medAL-*reader* as a learning tool, efforts were made to simply present the decision tree logic for individual diagnostic and syndromic branches of the algorithm. |
| Some medicines not available at health facilities due to stock-outs | Sometimes medicines recommended by national guidelines were not available | Provide alternative medicines for most conditions in case the recommended one is not available. |
| Misunderstanding of the labelling of some clinical elements | The labelling of some symptoms and signs were not well understood by the clinician | Modification of labelling of some elements, clarification provided in the information button, and translation to local language |
| Some clinical signs not measured, especially when patients are many | Many clinicians did not always measure required clinical signs (anthropometrics, temperature, respiratory rate) and could thus not continue with the algorithm | Provide options to not measure some clinical signs and rather estimate the values (with warning that this is sub-optimal) to limit clinicians being 'stuck', to discourage false information to be entered, and to provide mentorship to those not measuring these signs |
| No clear identification of symptoms and signs that always result in severe disease / referral | Clinicians selected variables that resulted in a severe diagnosis, parenteral antibiotics, and referral, for which the clinician did not agree with. | Elements that result in the diagnosis of a severe disease and referral are highlighted |

decision logic for young infants less than 2 months, and in some countries decision logic for children 5–15 years old, and expanded clinical content for diagnoses not included in IMCI. It is now being further validated in several large clinical studies. Following established development protocols, attempts were made to ensure a transparent development process, multistakeholder collaboration, and end-user feedback [21,22,55,56]. Specifically, aligning the development process of ePOCT+ and specifications of medAL-*reader* to the requirements of the Target Product Profile for CDSAs was helpful to better meet the needs of end users in terms of quality, safety, performance and operational functionality [21]. The development of medAL-*creator*, allows non-IT specialists to be able to program the clinical algorithms using a no-code, drag and drop interface, a novel solution that democratizes the development of CDSAs. This is a big advantage when compared to other CDSA tools that generally require advanced IT knowledge to review and program the code of the CDSA. Nonetheless, there are several limitations and challenges with the development process and the end-result of ePOCT+ and the medAL-*suite*, for which ongoing modifications and improvements will be required.

First, while efforts were made to improve the performance of the algorithm, there was often a reliance on clinical guidelines which may not always be founded on the best/latest/highest quality evidence, or applicable to low resource primary care settings [57,58]. Furthermore, they require significant interpretation to transform into algorithms. Digital Adaptation Kits (DAKs) to guide implementers in how to interpret narrative guidelines to transform into digital platforms are currently being developed by the World Health Organization and should help address this challenge in the future [50,59]. Often supplementary evidence was needed to complement national and international guidelines. This evidence should ideally be identified through systematic reviews [60], however those are not always feasible. Leveraging existing evidence databases as done by another CDSA may be a more feasible method to avoid biases in identifying supporting evidence [61]. Among the supporting evidence identified, there was a paucity of evidence for conditions specific to older children above 5 years, prognostic studies in the primary care setting, and diagnostic studies for conditions other than serious bacterial infection and pneumonia. Evaluating the prognostic and diagnostic value of predictors and models used in ePOCT+ during the ongoing validation studies will help to develop more efficient and better performing algorithms optimised for the target population [50,62].

A number of considerations were taken into account when digitalizing and adapting paper guidelines. Among the most important considerations were the feasibility, acceptability, reliability, and diagnostic and prognostic performance of individual clinical elements, while also considering the overall performance of the algorithms in relation to the pre-test probability of the outcome or disease, and the clinician's overall impression. Often conflicts can arise among the various factors that must be considered, which leads to difficult decisions. For example the Delphi survey among Tanzanian health care workers found that capillary refill time may not be feasible in primary health settings, however it has been found to have good prognostic value [35]. Such difficult decisions were often taken with input from clinical experts from the country of implementation. Additional training on clinical signs deemed not feasible, could potentially allow for future modifications. Another difficult decision included the option of estimating results when measurements are not possible (e,g, respiratory rate). Health care workers often do not measure respiratory rate when following paper guidelines or using a CDSA [7,19]. If the CDSA does not allow the option of not being able to measure respiratory rate then health care workers may not be able to move forward using the tool, or may enter false data if indeed respiratory rate measurement is not feasible. Allowing health care workers to estimate the value is not ideal, but allows the health care worker to at the very least visually assess respiratory rate, and provide an input in order for the algorithm to reach a diagnosis.

This data can then be used to mentor health care workers that do not measure respiratory rate. Allowing clinicians to simply indicate that the respiratory rate was not possible to measure without forcing an estimation could be an option to consider, but would complicate the decision on what diagnosis to reach when selecting this option.

Many modifications to ePOCT+ and medAL-*reader* compared to previous generation CDSAs were implemented in order to help improve uptake, addressing previously shared concerns such as limited scope, and ease of use. medAL-*reader* was specifically designed to follow normal healthcare workflows, and incorporate more input from the healthcare workers. Compared to other CDSAs, medAL-*reader* includes new functions such as an emergency button, and the ability to accept or refuse a diagnosis or treatment. The introduction of other digital tools such as electronic medical records within the same health facilities creates challenges in uptake and may result in duplication of processes. As an example, it is estimated that there are over 160 digital health or health-related systems in Tanzania [63]. While efforts are currently being made to harmonize processes so that different digital systems can complement each other rather than creating additional work, this has not yet been achieved. It is important to note, that while ePOCT+ and medAL-*reader* may address some challenges to uptake of CDSAs, there are many extrinsic and intrinsic factors that are not addressed, such as the low perceived value of following guidelines, and lack of motivation partly related to poor remuneration [16,64].

The digitalization process allows for increased complexity in the algorithm compared to paper guidelines. However, this complexity may limit the understanding by healthcare workers. Understanding how a diagnosis and treatment plan is reached is fundamental to clinical and patient autonomy, important for continued learning, and for fostering trust in any algorithm.[65–67] Efforts were made to present simple decision tree logic for each diagnosis. Nevertheless, the optimal method of presentation of algorithm branches to assure understanding by primary healthcare workers should be further explored.

## Conclusion

ePOCT+ aims to improve clinical care of sick children in LMICs, notably by reducing unnecessary antibiotic prescription. We hope that the strong stakeholder involvement, the expanded scope of the clinical algorithm, and the novel software of the medAL-*suite* will result in high uptake, trust and acceptability. Widespread implementation will provide opportunities for dynamic and targeted refinements to the clinical content to improve the performance of the algorithm. We further hope that the easy-to-use platform of the medAL-*suite*, and the framework used to develop ePOCT+ will allow health authorities and local communities to be able to take ownership of ePOCT+ or their own clinical algorithm for future adaptations and developments. Future success however, is contingent on the harmonization with national health management information systems and other digital systems.

## Supporting information

**S1 Appendix. Prevalence of specific symptoms and diagnoses not covered in IMCI from Tanzania.**
(DOCX)

**S2 Appendix. Delphi survey on the reliability and feasibility of measurement of symptoms and signs.**
(DOCX)

**S3 Appendix. Prognostic value of predictors used in the ePOCT and ALMANACH electronic clinical decision support algorithms.**
(DOCX)

**S4 Appendix. Features of the medAL-*creator* and medAL-*reader software as defined by a clinical-IT collaboration with end-user feedback.***
(DOCX)

**S5 Appendix. Evaluation of ePOCT+ based on the characteristics set by the target product profile for electronic clinical decision support algorithm as defined by expert consensus.**
(DOCX)

## Acknowledgments

Emmanuel Barchichat, Alain Fresco, and Quentin Girard from Wavemind for the IT programming of medal-creator and medal-reader software. Martin Norris, Lisa Cleveley, Dr Sabine Renggli, Ibrahim Mtabene, Peter Agrea and Dr Godfrey Kavishe for the medAL-reader tests and suggestions for improvements to both medal-reader and medal-creator. Cecile Trottet for the statistical support. The many health care workers providing feedback on the tool, patients and caretakers involved with pilot and feasibility testing. Dr Arjun Chandna and Janet Urquhart for helpful comments on the manuscript.

## Author Contributions

**Conceptualization:** Rainer Tan, Kristina Keitel, Valérie D'Acremont.

**Formal analysis:** Rainer Tan, Josephine Van De Maat, Mary-Anne Hartley.

**Funding acquisition:** Valérie D'Acremont.

**Investigation:** Rainer Tan, Ludovico Cobuccio, Fenella Beynon, Gillian A. Levine, Nina Vaezipour, Lameck Bonaventure Luwanda, Chacha Mangu, Nahya Salim, Karim Manji, Helga Naburi, Lulu Chirande, Lena Matata, Method Bulongeleje, Robert Moshiro, Andolo Miheso, Peter Arimi, Ousmane Ndiaye, Moctar Faye, Aliou Thiongane, Shally Awasthi, Kovid Sharma, Gaurav Kumar, Josephine Van De Maat, Victor Rwandarwacu, Théophile Dusengumuremyi, John Baptist Nkuranga, Emmanuel Rusingiza, Lisine Tuyisenge, Mary-Anne Hartley, Kristina Keitel, Valérie D'Acremont.

**Methodology:** Rainer Tan, Ludovico Cobuccio, Fenella Beynon, Gillian A. Levine, Nina Vaezipour, Lameck Bonaventure Luwanda, Chacha Mangu, Alan Vonlanthen, Olga De Santis, Nahya Salim, Karim Manji, Helga Naburi, Lulu Chirande, Lena Matata, Method Bulongeleje, Robert Moshiro, Andolo Miheso, Peter Arimi, Ousmane Ndiaye, Moctar Faye, Aliou Thiongane, Shally Awasthi, Kovid Sharma, Gaurav Kumar, Josephine Van De Maat, Alexandra Kulinkina, Victor Rwandarwacu, Théophile Dusengumuremyi, John Baptist Nkuranga, Emmanuel Rusingiza, Lisine Tuyisenge, Mary-Anne Hartley, Kristina Keitel, Valérie D'Acremont.

**Project administration:** Alan Vonlanthen, Alexandra Kulinkina, Vincent Faivre, Julien Thabard, Valérie D'Acremont.

**Software:** Rainer Tan, Ludovico Cobuccio, Fenella Beynon, Gillian A. Levine, Lameck Bonaventure Luwanda, Chacha Mangu, Alan Vonlanthen, Olga De Santis, Nahya Salim, Karim Manji, Helga Naburi, Lulu Chirande, Lena Matata, Method Bulongeleje, Robert Moshiro, Andolo Miheso, Peter Arimi, Ousmane Ndiaye, Moctar Faye, Aliou Thiongane, Shally

Awasthi, Kovid Sharma, Gaurav Kumar, Alexandra Kulinkina, Victor Rwandarwacu, Théophile Dusengumuremyi, John Baptist Nkuranga, Emmanuel Rusingiza, Lisine Tuyisenge, Vincent Faivre, Julien Thabard, Kristina Keitel, Valérie D'Acremont.

**Supervision:** Kristina Keitel, Valérie D'Acremont.

**Visualization:** Rainer Tan.

**Writing – original draft:** Rainer Tan.

**Writing – review & editing:** Rainer Tan, Ludovico Cobuccio, Fenella Beynon, Gillian A. Levine, Nina Vaezipour, Lameck Bonaventure Luwanda, Chacha Mangu, Alan Vonlanthen, Olga De Santis, Nahya Salim, Karim Manji, Helga Naburi, Lulu Chirande, Lena Matata, Method Bulongeleje, Robert Moshiro, Andolo Miheso, Peter Arimi, Ousmane Ndiaye, Moctar Faye, Aliou Thiongane, Shally Awasthi, Kovid Sharma, Gaurav Kumar, Josephine Van De Maat, Alexandra Kulinkina, Victor Rwandarwacu, Théophile Dusengumuremyi, John Baptist Nkuranga, Emmanuel Rusingiza, Lisine Tuyisenge, Mary-Anne Hartley, Vincent Faivre, Julien Thabard, Kristina Keitel, Valérie D'Acremont.

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
