## [Decision Letter · Decision Letter 0]

12 Sep 2022

PDIG-D-22-00055

ePOCT+ and the medAL-*suite*: Development of an electronic clinical decision support algorithm and digital platform for paediatric outpatients in low- and middle-income countries

PLOS Digital Health

Dear Dr. Tan,

Thank you for submitting your manuscript to PLOS Digital Health. After careful consideration, we feel that it has merit but does not fully meet PLOS Digital Health's publication criteria as it currently stands. Therefore, we invite you to submit a revised version of the manuscript that addresses the points raised during the review process.

Please submit your revised manuscript within 30 days Oct 12 2022 11:59PM. If you will need more time than this to complete your revisions, please reply to this message or contact the journal office at digitalhealth@plos.org. Please include the following items when submitting your revised manuscript:

We look forward to receiving your revised manuscript.

Kind regards,

Ryan S McGinnis, Ph.D.

Academic Editor

PLOS Digital Health

Journal Requirements:

1. Please ensure that the funders and grant numbers match between the Financial Disclosure field and the Funding Information tab in your submission form. Note that the funders must be provided in the same order in both places as well.

2. Please update your online Competing Interests statement. If you have no competing interests to declare, please state: “The authors have declared that no competing interests exist.”

3. In the online submission form, you indicated that your data will be submitted to a repository upon acceptance. We strongly recommend all authors deposit their data before acceptance, as the process can be lengthy and hold up publication timelines. Please note that, though access restrictions are acceptable now, your entire data will need to be made freely accessible if your manuscript is accepted for publication. This policy applies to all data except where public deposition would breach compliance with the protocol approved by your research ethics board. If you are unable to adhere to our open data policy, please kindly revise your statement to explain your reasoning and we will seek the editor's input on an exemption. Please be assured that, once you have provided your new statement, the assessment of your exemption will not hold up the peer review process.

4. Please ensure that the Title in your manuscript and the Title in your online submission form are the same.

5. Please provide separate figure files in .tif or .eps format only and remove any figures embedded in your manuscript file. Please also ensure that all files are under our size limit of 10MB.

For more information about how to convert your figure files please see our guidelines: https://journals.plos.org/digitalhealth/s/figures

6. Some material included in your submission may be copyrighted. According to PLOS’s copyright policy, authors who use figures or other material (e.g., graphics, clipart, maps) from another author or copyright holder must demonstrate or obtain permission to publish this material under the Creative Commons Attribution 4.0 International (CC BY 4.0) License used by PLOS journals. Please closely review the details of PLOS’s copyright requirements here: PLOS Licenses and Copyright. If you need to request permissions from a copyright holder, you may use PLOS's Copyright Content Permission form.

Potential Copyright Issues:

Figures 1 and 2: Please confirm whether you drew the images / clip-art within the figure panels by hand. If you did not draw the images, please provide (a) a link to the source of the images or icons and their license / terms of use; or (b) written permission from the copyright holder to publish the images or icons under our CC-BY 4.0 license. Alternatively, you may replace the images with open source alternatives. See these open source resources you may use to replace images / clip-art:

- https://openclipart.org/

Additional Editor Comments (if provided):

Reviewers' comments:

Reviewer's Responses to Questions

**Comments to the Author**

1. Does this manuscript meet PLOS Digital Health’s publication criteria? Is the manuscript technically sound, and do the data support the conclusions? The manuscript must describe methodologically and ethically rigorous research with conclusions that are appropriately drawn based on the data presented.

Reviewer #1: Yes

Reviewer #2: Yes

2. Has the statistical analysis been performed appropriately and rigorously?

Reviewer #1: N/A

Reviewer #2: N/A

3. Have the authors made all data underlying the findings in their manuscript fully available (please refer to the Data Availability Statement at the start of the manuscript PDF file)?

Reviewer #1: Yes

Reviewer #2: Yes

4. Is the manuscript presented in an intelligible fashion and written in standard English?

Reviewer #1: Yes

Reviewer #2: Yes

5. Review Comments to the Author

Reviewer #1: This is a potentially useful paper for individuals interested in developing clinical decision support algorithms and other CDS tools. The general process of development is good, but some areas need more explanation to be useful to others. These areas are described below.

lines 178-180-- If a number of key variables are omitted how does that affect the validity of the algorithm? This needs more discussion

line 268-- Describe the more formal decision processes that were used.

lines 314-318-- More detail on the field studies and user testing experience should be included.

Reviewer #2: Thank you for the opportunity to review this manuscript, which describes development of a new generation CDSA ePOCT, with expanded age groups and perceived user-friendly interface, that has been implemented in a number of countries, and that is awaiting further validation in clinical studies. 

The manuscript is generally well written, albeit repetitive in some areas. The authors describe in detail the background process undertaken to improve the content of the CDSA, which was comprehensive and for which they should be congratulated, and they include a description of the feedback provided from users which was taken on to further improve the tool. 

The following are some comments for the authors to consider regarding their manuscript - 

Major - 

Abstract:

Suggest that it is important to highlight in the abstract that the algorithm is undergoing further validation in clinical studies.

Clinical Algorithm:

line 158 - authors state that ‘for conditions not covered in these guidelines’ - is it not clear what the process/reference was for identifying conditions not covered in these guidelines, could the authors please expand, so that readers understand the systematic approach taken, or is this entire paragraph repeating from paragraph commencing line 145 on page 7?

Feasibility, acceptability, and reliability of predictors:

The authors refer to the Delphi survey amongst 30 Tanzanian healthcare workers which is further described in the appendix, that lead to omission of widely recognised warning signs e.g. capillary refill time and cool peripheries — could authors consider commenting, perhaps in the discussion - about whether these signs are definitely not feasible in all settings that the CDSA is currently being implemented, and speculate on whether appropriate training should be implemented / what difference this would make to morbidity and mortality, based on other studies in similar settings (i.e. other widely implemented CDSAs)? Based on the author’s previous studies referenced in the supplementary appendix comparing ALMANACH with ePOCT, perhaps these signs are not highly important but I think it would add depth to the paper to discuss this element. Furthermore, could authors further describe or state the number of feasibility tests done on real and fictional cases that led to this omission in this section of the manuscript or supplementary appendix?

In the feasibility tests, authors do not comment on the training required to implement this CDSA in the target setting - i.e. how much time is required for primary healthcare workers to be trained to use this tool?

Table 1 - could authors consider expanding on the decision for some clinical signs that were not measured to be allowed to be an estimate rather than (or in addition to) a “not measured” option?

Discussion: 

Would the authors like to comment on how their CDSA compares to other widely used CDSAs in this setting if applicable

Minor - 

Figure 3 is difficult to read - font small and blurry in centre algorithm

6. PLOS authors have the option to publish the peer review history of their article (what does this mean?). If published, this will include your full peer review and any attached files.

**Do you want your identity to be public for this peer review?** For information about this choice, including consent withdrawal, please see our Privacy Policy.

Reviewer #1: No

Reviewer #2: No

---

## [Decision Letter · Decision Letter 1]

9 Nov 2022

PDIG-D-22-00055R1

ePOCT+ and the medAL-*suite*: Development of an electronic clinical decision support algorithm and digital platform for pediatric outpatients in low- and middle-income countries

PLOS Digital Health

Dear Dr. Tan,

Thank you for submitting your manuscript to PLOS Digital Health. After careful consideration, we feel that it has merit but does not fully meet PLOS Digital Health's publication criteria as it currently stands. Therefore, we invite you to submit a revised version of the manuscript that addresses the points raised during the review process.

Please submit your revised manuscript within 30 days Dec 09 2022 11:59PM. If you will need more time than this to complete your revisions, please reply to this message or contact the journal office at digitalhealth@plos.org. Please include the following items when submitting your revised manuscript:

We look forward to receiving your revised manuscript.

Kind regards,

Ryan S McGinnis, Ph.D.

Academic Editor

PLOS Digital Health

Journal Requirements:

2. Figures 1-3 contains screenshots. We are not permitted to publish these under our CC-BY 4.0 license; websites are usually intellectual property and are copyrighted.This includes peripheral graphics of the web browser such as the buttons. We ask that you please remove or replace it.

Additional Editor Comments (if provided):

Reviewers' comments:

Reviewer's Responses to Questions

**Comments to the Author**

1. If the authors have adequately addressed your comments raised in a previous round of review and you feel that this manuscript is now acceptable for publication, you may indicate that here to bypass the “Comments to the Author” section, enter your conflict of interest statement in the “Confidential to Editor” section, and submit your "Accept" recommendation.

Reviewer #1: (No Response)

Reviewer #2: All comments have been addressed

2. Does this manuscript meet PLOS Digital Health’s publication criteria? Is the manuscript technically sound, and do the data support the conclusions? The manuscript must describe methodologically and ethically rigorous research with conclusions that are appropriately drawn based on the data presented.

Reviewer #1: Yes

Reviewer #2: Yes

3. Has the statistical analysis been performed appropriately and rigorously?

Reviewer #1: Yes

Reviewer #2: N/A

4. Have the authors made all data underlying the findings in their manuscript fully available (please refer to the Data Availability Statement at the start of the manuscript PDF file)?

Reviewer #1: Yes

Reviewer #2: Yes

5. Is the manuscript presented in an intelligible fashion and written in standard English?

Reviewer #1: Yes

Reviewer #2: Yes

6. Review Comments to the Author

Reviewer #1: Most changes have been made, but a few still need to be addressed. I could not find the number of feasibility tests on real cases in the body of the manuscript. It did not seem to be where the authors said it was. In addition, the authors were asked to compare their system to other similar systems. Although it is too early for the actual results/performance to be compared, the design, intent, content of the system can and should be compared to other similar systems.

Reviewer #2: No further comments, thank you.

7. PLOS authors have the option to publish the peer review history of their article (what does this mean?). If published, this will include your full peer review and any attached files.

**Do you want your identity to be public for this peer review?** For information about this choice, including consent withdrawal, please see our Privacy Policy.

Reviewer #1: No

Reviewer #2: No

---

## [Editor Report · Decision Letter 2]

23 Nov 2022

ePOCT+ and the medAL-*suite*: Development of an electronic clinical decision support algorithm and digital platform for pediatric outpatients in low- and middle-income countries

PDIG-D-22-00055R2

Dear Dr Tan,

We are pleased to inform you that your manuscript 'ePOCT+ and the medAL-*suite*: Development of an electronic clinical decision support algorithm and digital platform for pediatric outpatients in low- and middle-income countries' has been provisionally accepted for publication in PLOS Digital Health.

Best regards,

Ryan S McGinnis, Ph.D.

Academic Editor

PLOS Digital Health